# Sowing Density Effects in Cotton Yields and Its Components

**Manuel Guzman [1,2,\*]**, **Luis Vilain [1]**, **Tatiana Rondon [2,3]** and **Juan Sanchez [1]**

[1]  Instituto Nacional de Investigaciones Agropecuarias (INIA), km 5, Carretera Nacional Acarigua–Barquisimeto, 3303 Araure, Venezuela

[2]  Corporación Colombiana de Investigación Agropecuaria (AGROSAVIA), Centro de Investigación La Selva, km 7, vía Rionegro-Las Palmas, Vereda Llanogrande, 054048 Rionegro, Colombia

[3]  Facultad de Agronomía, UCV-Campus Maracay, Universidad Central de Venezuela, Av. Universidad, vía El Limón, 2101A Maracay, Venezuela

\*  Correspondence: maguzman@agrosavia.co; Tel.: +57-319-610-8802

**Abstract:** Evaluation of sowing density is an important factor for achieving maximum yields without affecting other agronomic traits. Field experiments were conducted during three consecutive years (2008, 2009 and 2010) to determinate the effect of four sowing density (62,500; 83,333; 100,000 and 142,857 pl ha$^{-1}$) on yields and its components of two cotton varieties, 'Delta Pine 16' and 'SN-290' in Venezuela. The traits evaluated were lint yield, boll weight, number of seeds per boll, 100-seed weight, and fiber content. Highly significant differences ($p \leq 0.01$) were observed among genotypes, sowing density and their interactions for all traits. Sowing density was not affected by year factor. High lint yield was found in 'SN-290' (4216.2 kg ha$^{-1}$) at 100,000 pl ha$^{-1}$; and in 'Delta Pine 16' (3917.3 kg ha$^{-1}$) at 83,333 pl ha$^{-1}$. The highest sowing density (142,857 pl ha$^{-1}$), decrease lint yield and yield components in the genotypes. The highest boll weight was obtained by 'SN-290' with 6.4 g in average. All sowing densities evaluated resulted in lint percentages above 40%. Cotton lint yield was positively correlated with all yield components. Our results indicate that highest lint yields could be obtained with sowing densities between 83,333 and 100,000 pl ha$^{-1}$ depending upon varieties used across savannahs of Venezuela.

**Keywords:** *Gossypium hirsutum* L.; lint yield; planting density; row spacing; spatial arrangement

## 1. Introduction

Cotton (*Gossypium hirsutum* L.) is the most important natural fiber crop worldwide, with 33 million hectares cultivated in 82 countries [1], mainly in Asia and America. In 2017, the cotton yield in Venezuela was 1085 kg ha$^{-1}$, which is very low compared with yields worldwide (2254 kg ha$^{-1}$) [2]. However, in Venezuela the main potential of the crop is attributed to the quality of fiber produced in terms of length, strength and micronaire parameters [3], obtained from medium fiber varieties which are the most widespread in the country. The patterns of cotton production have not changed in the last three decades in Venezuela. The sowing systems correspond to (i) hand-planting in floodplain, located in the south of Guarico state and north of Apure state, in areas adjacent to the Orinoco and Apure rivers, respectively [4]; and (ii) mechanized in savannas with poorly (Portuguesa and Barinas states) and well (Monagas and Anzoategui states) drained soils. The mechanized planting system is more successful due to varieties' homogeneity in size, precocity and productivity, facilitating mechanized harvesting.

In Venezuela, different studies have focused on the identification of foreign varieties or those developed in the national cotton breeding program of INIA (Instituto Nacional de Investigaciones Agropecuarias) that are adapted to specific edaphic and climatic conditions, maximizing expression of

fiber yield and quality [5–7]. These studies have been developed based on spatial arrangement less than or equal to 62,500 pl ha$^{-1}$. However, no further efforts have been made to evaluate a greater number of plants per unit area and its impact on agronomic traits of interest at agroecologies in Venezuela.

Yield and its components are influenced by genetic parameters and agronomic practices, where sowing density plays an important role [8]. Optimal sowing density can vary between regions, then it is necessary to conduct studies in similar areas in management, soils and weather patterns. In the USA, the plant arrangement used is very different among regions in order to maximize yields; i.e., 12.6 pl m$^{-2}$ in Georgia [9], 15.3 pl m$^{-2}$ in Louisiana [10], 6.6 pl m$^{-2}$ in Mississippi [11], and 10.0 pl m$^{-2}$ in Arizona [12]. On the other hand, plant densities in China are between 22.5 pl m$^{-2}$ in the northwest [13] and 2.0 pl m$^{-2}$ in the Yangtze river valley [14]. These differences are based on the architecture of the cotton genotype, agronomic management, and harvesting practices.

The generation of new varieties and alternative agronomic practices have encouraged changes in the production systems of crops [15] with emphasis in manipulation of row spacing dimensions and plant populations. Several studies in cotton report yield increases and variations in fiber quality due to changes in the spatial distribution of plants [9,16,17] and is used as an alternative on commercial fields. Briggs et al. [18] introduced the concept of ultra-narrow rows in cotton, which became very popular worldwide. However, the use of high planting densities increases the appearance of diseases, smaller bolls, shading of immature flower, delays in maturation and reduction in plant size [19,20]. Many of these limitations have already been overcome through breeding. Kerby et al. [21] reported that the increase in plant density from 10 to 15 plants by m$^{-2}$ delayed maturity in undetermined growth habits of cotton genotypes, while those with determined growth habit were not affected by this variation. Traditionally, the distance between single rows of cotton has varied between 80 and 100 cm [22]; however, since 1990 the use of ultra-narrow rows (25 cm or less) was increased as an alternative to reduce production costs and increase yields [23], with the limitation of mechanized harvesting causing significant losses in the field and during ginning.

Wilson et al. [24] reported the use of rows spaced at 38 cm, obtaining yields equal to or higher than those obtained with rows between 97 and 102 cm. Stephenson and Brecke [25] reported slight increases in yield when planting double rows at 19 cm, compared with simple rows at 76 cm. On the other hand, Reddy et al. [22] did not find significant differences in performance when experimenting with double and simple rows at 25 and 102 cm, respectively.

There is not information on the effect of sowing densities on cotton yield performance in Venezuela. Therefore, the objective of this study was to evaluate the effect of different sowing densities on yield and other agronomic variables of interest in two commercial varieties of cotton 'Delta Pine 16' and 'SN-290', in order to identify technological alternatives to make efficient use of land and increase the productivity and profitability of cotton in Venezuela.

## 2. Materials and Methods

### 2.1. Experimental Site

The experiments were conducted during the cotton growing season (July–November) during three years (2008, 2009 and 2010) at the experimental field of INIA-Portuguesa, located in Araure, Portuguesa state, Venezuela (9°36'51" N, 69°14'34" W, 233 m a.s.l). All the field experiments were sown in mid-July in each year. The soil taxonomy is an Entisol, Aeric Tropic Flavaquent, silt, mixed, non-acid, isohyperthermic. The climatic classification according to Köppen is Tropical Dry (Aw). During the trials, climatic data were collected from an automatic weather station at a 2 km distance and summarized in monthly data. Historical weather data was obtained of the same station. Across years, the rainfall accumulation was of 879 mm and temperatures fluctuated between 22.1 and 29.3 °C on average during the field trials (Figure 1).

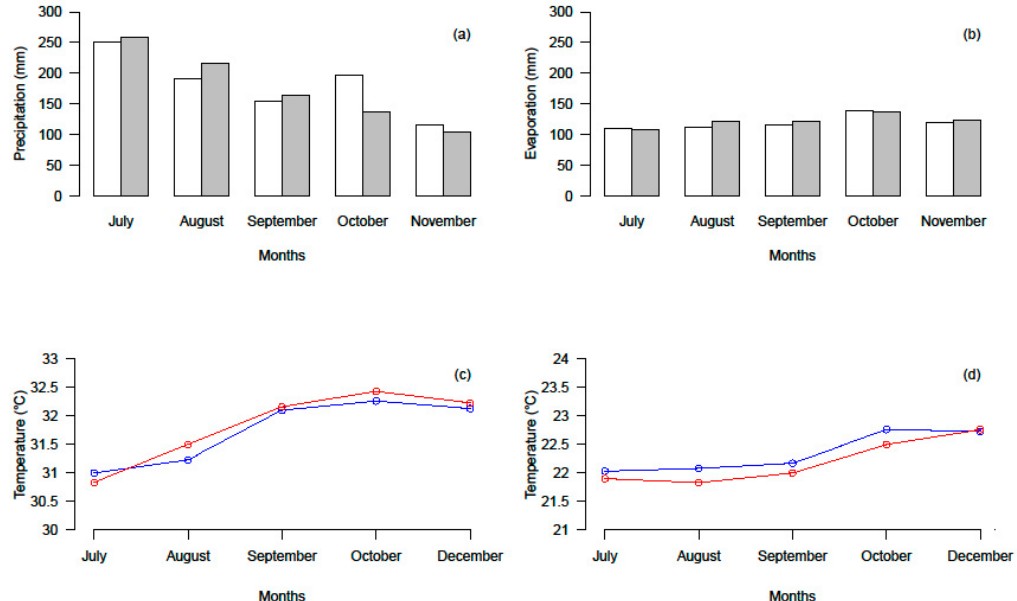

**Figure 1.** Average monthly weather data at Araure, Venezuela during cotton growing season (July-November) of historical period 1988–2008 (white bars and blue lines) and trial period 2008–2010 (grey bars and red lines): (**a**) precipitation and (**b**) evaporation expressed in mm; (**c**) maximum and (**d**) minimum temperature expressed in °C.

## 2.2. Plant Material, Treatments and Experimental Design

The genotypes evaluated were the commercial varieties 'Delta Pine 16' and 'SN-290', genotypes with medium fiber and widely accepted by farmers in Venezuela, because of their fiber quality. In total, four spatial arrangements were evaluated, only varying plants' distance between rows. Three simple row treatments spaced at 50, 60 and 80 cm apart; and a double row treatment, spaced 30 cm between the double row and 80 cm between single rows (Figure 2). The final spatial arrangements were 100,000; 83,333; 62,500 and 142,857 pl ha$^{-1}$. The treatments were arranged as split plots in a completely randomized block design with three replications at each site. The main plot was represented by the spatial arrangements, and the subplots by the varieties. The experimental unit was four row plots that were 15 m long.

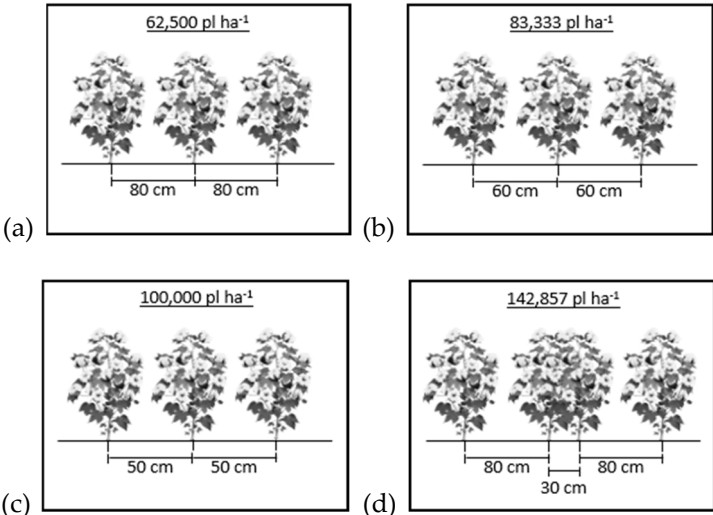

**Figure 2.** Schematic representation of cotton sowing densities evaluated in this study. Sowing densities of (**a**) 62,500; (**b**) 83,333; (**c**) 100,000 and (**d**) 142,857 pl ha$^{-1}$ are shown.

### 2.3. Management of Trials and Data Collection

Planting was done manually, putting two seeds 0.2 m apart per hole, and 15–18 days after sowing (DAS) thinned to one plant. Plants were established from seeds protected with fungicide (Thiophanate-methyl). The emergence of cotton seedlings was homogeneous at 7 DAS approximately. During land preparation at each year, the trials received the recommended fertilization rates according to soil analysis, 52 kg ha$^{-1}$ of N, 90 kg ha$^{-1}$ de $P_2O_5$ and 50 kg ha$^{-1}$ of $K_2O$. Second doses of N (80 kg ha$^{-1}$) were side-dressed at 35 days after crop emergence. Nonlimiting growth conditions were maintained throughout the experiment, while weed control and insect control were done according to infestation levels. In the three years, a pre-emergence herbicide, Prowl®400 (pendimenthalin; 3.5 L ha$^{-1}$), was applied at planting. During cotton growth, larvae of fall armyworm *Spodoptera frugiperda* Smith (Lepidoptera: Noctuidae) and adults of boll weevil *Anthonomus grandis* Boheman (Coleoptera: Curculionidae) were controlled using Engeo® SC (lambda-cyhalothrin and thiamethoxam; 0.3 l.ha$^{-1}$) and Decis® EC (deltamethrin; 0.4 l.ha$^{-1}$). The entire rows per plot were harvested by hand to determine lint yield (LY, kg ha$^{-1}$). A subsample of 100 randomly bolls were collected from each plot, to determine yield components including boll weight (BW, g), seed per boll (SB), seed index (SI, g) from the weight of 100 seeds, and lint content (LP, %) was as follows:

$$LY = \frac{Fibre\ weight(g)}{Fibre\ weight(g) + Seed\ weight(g)} \times 100 \tag{1}$$

Seed cotton samples were ginned on a laboratory-scale gin (TB510A, TESTEX, Dongguan, GD, China) to separate lint from seeds.

### 2.4. Statistical Analysis

Statistical analyses were carried out for the average of each plot, per year. Means were compared across years using generalized lineal models (GLM) implemented in SAS 9.3 statistical software (SAS Institute Inc., Cary, NC, USA, 2011). The Shapiro–Wilk statistic test indicated normality for all data. When significant differences were detected among treatments at each year, a combined analysis was conducted, with varieties and spatial arrangement as fixed effects and years as random effects. Means were separated using Tukey test at $p \leq 0.05$. Correlation analysis by Pearson was used to determine the relationship between each pair of traits evaluated.

## 3. Results and Discussion

Homogeneity of variance test indicated homogeneous error for each trait in the three years and allowed for a combined analysis across years. The combined analysis of variance showed that sowing density had highly significance effects ($p \leq 0.01$) on lint yield, boll weight, lint percent, number of seeds per boll and seed index (Table 1). Similarly, mean squares due to genotypes and the interaction of sowing densities with genotypes had significant effects in all variables evaluated except for lint percentage in the interaction term, for which the value was non-significant.

Mean squares indicated variability among cotton performance at different sowing densities due to their diverse genetic background and weather conditions across years. Low values for the coefficients of variation (15% or less) for all evaluated traits indicated high experimental precision. In this study, the year effect was significant, as expected for the typical weather variation in the lowlands and savannahs of Venezuela, where the rainfall distribution is critical in the rainy season. Precipitation at the beginning of the experiments contributes with cotton growth and lint yields until the crop is relatively advanced. This crop requires abundant water during the vegetative stage in order to complete all physiological processes, while it is necessary a low air humidity and no rain during the maturity stage to avoid productivity losses.

**Table 1.** Mean squares and significant test for yield and yield components of sowing density evaluated during 2008 to 2010.

| Source of Variation | df | LY (kg ha$^{-1}$) | | BW (g) | | LP (%) | | SB (boll$^{-1}$) | | SI (g) | |
|---|---|---|---|---|---|---|---|---|---|---|---|
| Replication (Rep) | 2 | 15,579 | | 0.0208 | | 0.1539 | | 0.1148 | | 0.2228 | |
| Sowing density (Trat) | 3 | 3,716,671 | *** | 1.5745 | *** | 3.4216 | *** | 8.4482 | *** | 3.8261 | *** |
| Genotype (Gen) | 1 | 1,876,630 | *** | 0.9090 | *** | 1.6260 | *** | 5.1040 | *** | 1.8625 | *** |
| Year | 2 | 421,222 | * | 0.1618 | * | 0.4648 | ** | 1.4109 | ** | 0.5605 | ** |
| Trat × Gen | 3 | 97,700 | * | 0.1751 | * | 0.1841 | | 0.8480 | * | 0.2938 | * |
| Trat × Year | 6 | 114,712 | | 0.0654 | | 0.0925 | | 0.2746 | | 0.0943 | |
| Gen × Year | 2 | 269,942 | | 0.0809 | | 0.1595 | | 0.4777 | | 0.1624 | |
| Trat × Gen × Year | 6 | 288,374 | * | 0.1370 | * | 0.1878 | | 0.4233 | | 0.2257 | |
| Error | 46 | 104,976 | | 0.0441 | | 0.1012 | | 0.2566 | | 0.1059 | |
| Grand Mean | | 3659.3 | | 6.31 | | 41.25 | | 24.74 | | 10.05 | |
| Minimum mean | | 2349.0 | | 5.20 | | 40.11 | | 23.15 | | 9.02 | |
| Maximum mean | | 4625.0 | | 7.06 | | 42.31 | | 26.90 | | 10.98 | |
| SD | | 563.87 | | 0.37 | | 0.53 | | 0.86 | | 0.56 | |
| CV (%) | | 15.41 | | 3.33 | | 0.77 | | 2.05 | | 3.24 | |

*, ** and *** significant at $p \leq 0.05$; $p \leq 0.01$ and $p \leq 0.001$, respectively. df, degrees of freedom; LY, lint yield; BW, boll weight; LP, lint percentage; SB, number of seeds per boll; SI, seed index.

For each of the traits evaluated, the percentage of sums of squares (SS) remaining (for year, genotype, sowing density and genotype-sowing density interaction) ranged between 65 and 68% (Table 2). For the sowing density component, high percentages of SS were found for seed per boll (77%) and for lint yield (76%). On the other hand, genotype component values were 14% or less for all traits. Genotype by sowing density and years effects were lower than other SS sources, ranging from 4 to 8%. Seed index showed high percentages at each source of variation evaluated.

**Table 2.** Portion of sums of squares (SS) attributed to year, genotype (Gen), sowing density (Trat) and genotype × sowing density interaction as a percentage of the total sums of squares remaining after removing sums of squares due to replication, Trat × Year, Gen × Year, and Trat × Gen × Year.

| | LY (kg ha$^{-1}$) | BW (g) | LP (%) | SB (boll$^{-1}$) | SI (g) |
|---|---|---|---|---|---|
| Pooled error | 35 | 35 | 32 | 33 | 32 |
| Remaining | 65 | 65 | 68 | 67 | 68 |
| Year | 6 | 5 | 7 | 7 | 8 |
| Genotype (Gen) | 13 | 14 | 12 | 12 | 14 |
| Sowing Density (Trat) | 76 | 73 | 75 | 77 | 71 |
| Gen × Trat | 6 | 8 | 6 | 4 | 7 |

LY, lint yield; BW, boll weight; LP, lint percentage; SB, number of seeds per boll; SI, seed index.

Interactions were observed between sowing density and genotype for all traits evaluated and is shown in Figure 3; Figure 4. The typical sowing density in Venezuela is 62,500 pl ha$^{-1}$. However, an increased in final population was favorable for all evaluated traits in both genotypes, except when density was 142,857 pl ha$^{-1}$, where values decrease significantly.

For all treatments, the lint yield (LY) average was 3,659.3 kg ha$^{-1}$. 'SN-290' had high lint yields (4,216.2 kg ha$^{-1}$) at 100,000 pl ha$^{-1}$, representing an increase of 15.2% over the average. In general, across different sowing densities, 'SN-290' had a better lint yield than 'Delta Pine 16', with 3820.8 and 3497.9 kg ha$^{-1}$, respectively (Figure 3a). For each genotype, the best lint yields were obtained with 83,333 pl ha$^{-1}$ for 'Delta Pine 16' and with 100,000 pl ha$^{-1}$ for 'SN-290', representing an increase in lint yield of 12 and 10.3%, respectively, according to average by genotype (Figure 3b). This study showed that sowing density and genotype influenced lint yield, boll weight, seed per boll, seed index and lint percentage. Increasing sowing density positively influenced lint yield and yield components, except for the sowing density of 142,857 pl ha$^{-1}$ that resulted in a lint yield reduction of 17%. Similar results were reported by Enriquez-Sanchez et al. [26] with the variety 'Delta Pine 5409' sowed at a density of 150,000 pl ha$^{-1}$ in Mexico, using subsurface drip irrigation. Yield per unit area generally increases

with high plant density. Although, as plant density is increased yield per unit area will approach an upper limit, plateau, and then decline because of competition for resources such as light, water and nutrients among plants. Higher sowing density may cause an overlap canopy and shading of lower leaves in the canopy according to the architecture of the plant evaluated. In this study, both 'SN-290' and 'Delta Pine 16' are bush type with longer fruiting branches, shorter stature, wider canopy and more vegetative branches, and an excessive overlap could cause a decrease in the plant metabolism. A rational plant density provides a better canopy micro-environment to gain higher yield [19] and reduce inputs by minimizing seed use without sacrificing yields [17].

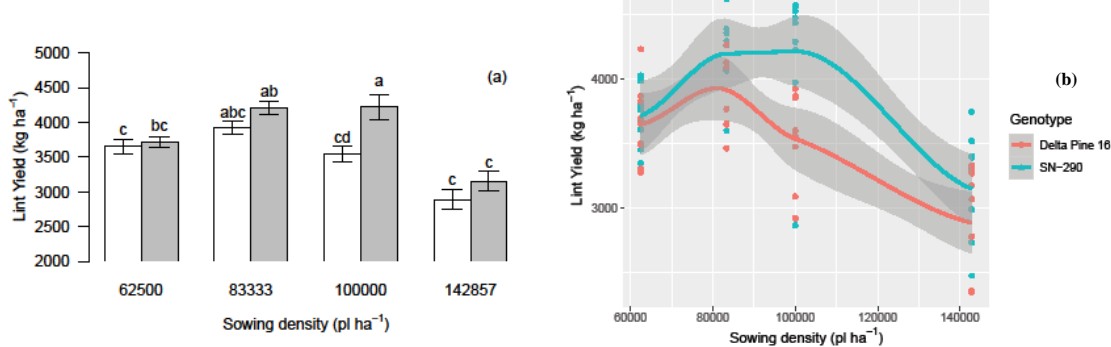

**Figure 3.** (**a**) Sowing density effects on cotton lint yield and (**b**) 95% confidence intervals for mean values. Each bar represents the mean (±SE) values by genotypes such as Delta Pine 16 (white bars) and SN-290 (grey bars). Bars sharing same letter do not differ significantly at $p \leq 0.05$ by Tukey's post-hoc test.

Boll weight average was 6.3 g, with 'SN-290' at 83,333 and 100,000 pl ha$^{-1}$ exhibiting the maximum values, 6% superior than the average (Figure 4a). Also, 'Delta Pine 16' at 83,333 pl ha$^{-1}$ had a higher BW value than the average. In this study, BW was directly related with LY. An increased in sowing densities at 142,857 pl ha$^{-1}$ resulted in fewer bolls at upper nodes and low weights, caused by the low number of main-stem nodes per plant, as reported by Clawson et al. [27] and McCarty et al. [28]. In contrast, Yang et al. [19], Zhi et al. [29] and Khan et al. [30] also reported a BW decreased as plant density increased. These authors indicated that at higher sowing density, cotton yield increased by an inverse relation between boll number and boll weight.

Figure 4b shows the lint percentage. This trait is important to cotton farmers because it represents the useful fraction used by the textile company. LP average was 41.3%, with 'SN-290' at 100,000 pl ha$^{-1}$ revealing a maximum value with of 41.73%. For both genotypes, LP values decreased significantly at 142,857 pl ha$^{-1}$, however LP values were not more inferior than 38%, which is the textile industry requirement in Venezuela. In this study, LP values were high, as compared with other studies carried out in similar environments [6,7]. In these cases plants were planted at 62,500 pl ha$^{-1}$ and followed the same pattern as reported by McCarty et al. [31], where LP become higher as plant population increased. Zhi et al. [29] reported an increase of 3.1% in LP in plants at 87,000 pl ha$^{-1}$ compare with plants at 15,000 pl ha$^{-1}$.

Seed per boll (SB) was very similar in both genotypes, with differences of up to 2.05 seeds per boll (Figure 4c). SI average was 10.1 g with maximum values of 10.62 g for 'SN-290' at 83.333 pl ha$^{-1}$ (Figure 4d). A greater number and weight of seeds per boll is desirable because of the greater surface area for lint production within each boll [32,33].

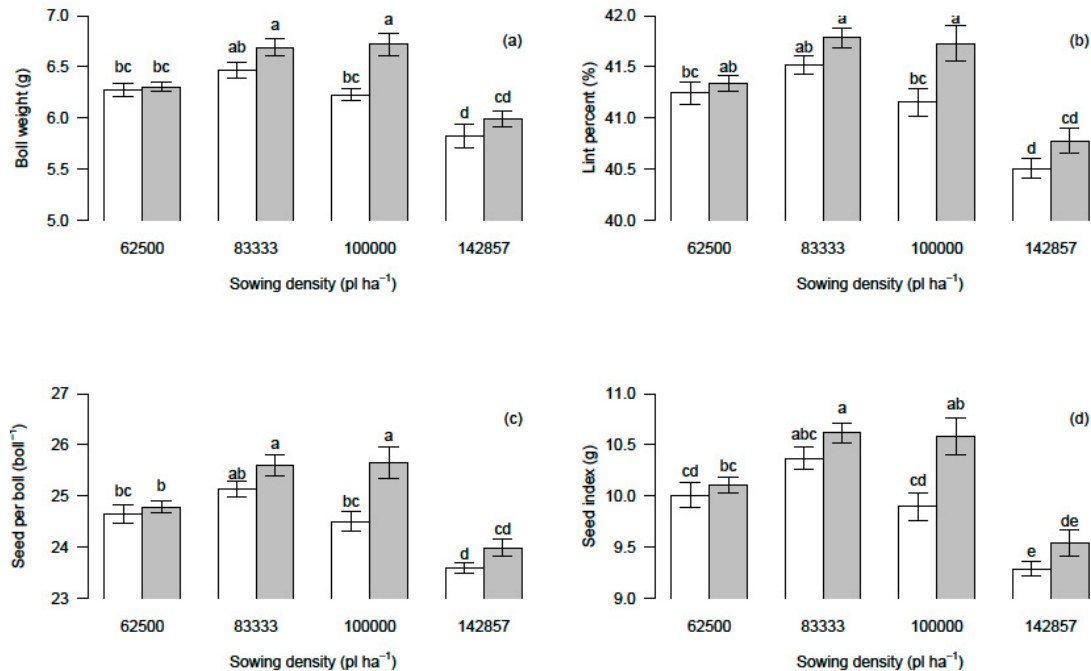

**Figure 4.** Sowing density effects on cotton yield components: (**a**) boll weight; (**b**) lint percent; (**c**) seed per boll; (**d**) seed index. Each bar represents the mean (±SE) values by genotypes, such as Delta Pine 16 (white) and SN-290 (grey). Bars sharing same letter do not differ significantly at $p \leq 0.05$ by Tukey's post-hoc test.

Results from correlation analysis of lint yield and other yields components are presented in Table 3. Lint yield was highly positively correlated with boll weight ($r = 0.98$), seed per boll ($r = 0.72$) and lint percentage ($r = 0.89$), indicating that dry matter accumulation is determinant. There were not significant correlations between seed per boll and boll weight ($r = 0.41$), seed per boll and lint percentage ($r = 0.36$) or seed index and lint percentage ($r = 0.33$). These yield components were not consistent and varied widely among sowing densities and genotypes.

**Table 3.** Pearson's correlation coefficients of lint yield and yield components of two cotton genotypes at four sowing densities.

|  | LY | BW | SB | SI |
|---|---|---|---|---|
| **BW** | 0.98** | | | |
| **SB** | 0.72** | 0.41 | | |
| **SI** | 0.45* | 0.65* | 0.88** | |
| **LP** | 0.89** | 0.96** | 0. 36 | 0.33 |

\* and ** significance at $p \leq 0.05$ and $p \leq 0.01$, respectively.

The superior performance of 'SN-290' over 'Delta Pine 16' in all evaluated traits, may be due to differences in the genetic backgrounds. 'Delta Pine 16' was an introduced from USA in the mid-1960s, meanwhile 'SN-290' is derived from a selection of different recombinant crosses between newest cotton genotypes from the 1985–1995 period, used as criterion for selection compact plants to improve light interception. Compact plants allow to increase the number of plants per area, without affecting other traits. Bednarz et al. [32] indicated relevant changes within-boll yield components during the last 60 years, where lint percentage has increased from 30 to 40% in average. These changes are attributed in part to selection of plants with different morphology that make it suitable for mechanical harvesting. However, high sowing densities are sensitive to fluctuations in the environmental conditions (such as micro-environments), compared with low densities ($\leq$62,000 pl ha$^{-1}$) [34]. These results shown that lint yield and yield components respond different to sowing densities and genotypes.

## 4. Conclusions

This study reveals that cotton lint yield is influenced by sowing density and genotype. Increasing the number of plants by closing rows can help obtaining higher lint yields. Across genotypes, boll weight, seed per boll, seed index and lint percentage increase linearly with sowing density up to 100,000 pl ha$^{-1}$. Sowing density of 142,857 pl ha$^{-1}$ causes a significant decreased of lint yield and other lint components in both examined genotypes. In general, 'SN-290' respond better than 'DeltaPine 16' to changes in sowing density. Therefore, this study suggests that farmers can improve their lint yield by increasing the sowing density from 62,500 to 83,333 and 100,000 pl ha$^{-1}$ for 'Delta Pine 16' and 'SN-290', respectively. The finding offers an alternative to cotton growers, who conventionally use wider rows and densities up to 62,500 pl ha$^{-1}$.

**Author Contributions:** Conceptualization, M.G. and L.V.; methodology, M.G., L.V and T.R.; formal analysis, M.G. and T.R.; investigation, M.G., T.R. and J.S.; data curation, J.S. and M.G.; writing—original draft preparation, M.G. and T.R.; writing—review and editing, M.G:, L.V., T.R: and J.S.; supervision, L.V.

**Funding:** This research was funded by the Ministry of Science and Technology of Venezuela, through its National Seed Program and Cotton Breeding Program. We would like to thank them for supporting this research.

**Conflicts of Interest:** The authors declare no conflict of interest. The funding sponsors had no role in the design of the study; in the collection, analyses, or interpretation of data; in the writing of the manuscript; and in the decision to publish the results.

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
