# Peer review of "Sowing Density Effects in Cotton Yields and Its Components"

_agronomy, doi:10.3390/agronomy9070349_

Round 1
Reviewer 1 Report
Please mention the years when the field trials had been carried out even in the Abstract.
Presenting a map about the experimental area would be beneficial in the M&M chapter.
When authors displaying the climatic conditions of the study site please compare the measured values in the growing seasons to a long term average because that way the weather could be characterised exactly.
In the M&M chapter mention the numbers of replication need to be indicated.
Please specify the name and active ingredients of the applied herbicides and insecticides.
Author Response
Point 1: Please mention the years when the field trials had been carried out even in the Abstract.
Response 1: Accepted. The information was added in the abstract.
Point 2: Presenting a map about the experimental area would be beneficial in the M&M chapter.
Response 2: The authors consider that it is not necessary to place a map in this section, understanding that the location and geographic coordinates were indicated in the text. In addition, the M&M section already has two figures (climatic data and a detail representation of sowing densities).
Point 3: When authors displaying the climatic conditions of the study site please compare the measured values in the growing seasons to a long term average because that way the weather could be characterised exactly.
Response 3: Accepted. The information was added in M&M section. A new figure was added with more specific information, including the variables: precipitation, evaporation, minimum and maximum temperatures recorded during a long-term period (1998-2008) and field trials period (2008-2010).
Point 4: In the M&M chapter mention the numbers of replication need to be indicated.
Response 4: Accepted. The information was added in M&M section.
Point 5: Please specify the name and active ingredients of the applied herbicides and insecticides.
Response 5: Accepted. The information was added in M&M section.
Reviewer 2 Report
This manuscript presents the result obtained in field experiments of four sowing density on yields and its components of two cotton varieties during three consecutive years. The study is interesting and with applicability. It is well written and I have only minor revisions to suggest:
- Figure 1 shows the average of monthly rainfall and temperature. How were they collected? Once a month, daily…??? The authors must specify it.
- I think that the way to present some results is not the most appropriate and make the article very hard. For example, the results of Table 3 could be presented graphically.
Author Response
Point 1: Figure 1 shows the average of monthly rainfall and temperature. How were they collected? Once a month, daily…??? The authors must specify it.
Response 1: Accepted. The information was added in M&M section. A new figure was added with more specific information, including the variables: precipitation, evaporation, minimum and maximum temperatures recorded during a long-term period (1998-2008) and field trials period (2008-2010).
Point 2: I think that the way to present some results is not the most appropriate and make the article very hard. For example, the results of Table 3 could be presented graphically.
Response 2: Accepted. Table 3 was changed by a figure, making the information presented easily understood by the reader. The calls to the figure were included in the text.